# Sexual and reproductive health service utilization of young girls in rural Ethiopia: What are the roles of health extension workers? Community-based cross-sectional study

Meskerem Jisso ,[1] Merga Belina Feyasa ,[2] Girmay Medhin,[3,4] Tegene Legese Dadi ,[1,4] Yilkal Simachew,[1] Bisrat Denberu,[5] Mulusew Gerbaba Jebena,[6] Yibeltal Kiflie Alemayehun,[4,6] Alula M Teklu[4]

For numbered affiliations see end of article.

**Correspondence to**
Meskerem Jisso;
meskyj@gmail.com

## ABSTRACT

**Objective** Little is known about the extent to which Health Extension Programme (HEP) has played its role to increase service uptake among young girls. This study aims to estimate the status of young girls' sexual and reproductive health (SRH) services utilisation in rural Ethiopia and to examine the role of health extension workers (HEWs) in this regard.

**Design** A cross-sectional study.

**Setting** A community-based study among all nine regions of Ethiopia.

**Participants** Nine hundred and two young girls aged 15–24 years were included in this study.

**Method** We used data from the national HEP assessment, collected from March to May 2019. Multilevel binary logistic regression was used to investigate the association between exposure to HEP and SRH services utilisation of young girls and we reported an adjusted OR with a corresponding 95% CI as measure of the degree of associations.

**Result** Only 19.18% (95% CI 16.74% to 21.89%) of young girls used SRH services with significant regional variability (intraclass correlation coefficien=17.16%; 95% CI 6.30% to 39.99%). Exposure to HEP (adjusted OR, aOR 3.13, 95% CI 2.03 to 4.85), knowing about the availability of HEP services (aOR 3.06, 95% CI 1.75 to 5.33) and having good trust in HEWs (aOR 1.82, 95% CI 1.07 to 3.10) and other sociodemographic factors were significantly associated with increased SRH services utilisation.

**Outcome** SRH service utilisation.

**Conclusion** Although the overall SRH service utilization of young girls in rural Ethiopia was very low, HEWs have a great contribution to improving service utilization of young girls through strong health education provided during home visits, school visits and at health posts. More investment along this line has the potential to improve service uptake among young girls. Encouraging HEWs to build trust among this segment of the population and creating awareness of SRH-related services is crucial to improv service uptake.

## STRENGTHS AND LIMITATIONS OF THIS STUDY

⇒ The study was done at country level, which is more representative.

⇒ It did not assess the attitude of health extension workers about sexual and reproductive health (SRH) delivery.

⇒ Recall bias that may have underestimated the magnitude of SRH services.

⇒ We could not assess the effect of cultural factors as a barrier for using SRH service qualitatively.

90% of them living in low-income and middle-income countries.[1] In sub-Saharan Africa, this segment of the population constitutes 23% of the region's population.[2] In Ethiopia, 34% of the total population is under 25 years and the majority of them (79% males and 78% females) reside in rural areas.[3] The number of young people in the least developed countries is expected to rise from 207 million in 2019 to 336 million by the year 2050.[4]

Having healthy youth leads to improved outcomes for a family, society and a nation as a whole. Hence, investing in their health has the potential to break the cycle of poverty through increased productivity, decreased health costs and improved social capital.[5] Yet young girls are undergoing great physical, physiological and emotional changes, which aggravate their vulnerability to sexual and reproductive health (SRH) problems.[5] Globally, in 2018, 1 in 25 young girls was at risk of early pregnancy, which is the leading cause of mortality among young girls.[6] Globally, three million young girls undergo an unsafe abortion, 460 000 were newly infected with HIV and 333 million of them were infected with a sexually transmitted infection

## BACKGROUND

Youths between 15 and 24 years of age account for more than 1.2 billion people globally with

(STI).[7] Similarly, in Ethiopia by 2016, 0.4% of youths were infected by HIV/AIDS,[8] and among university and college students, based on a 2019 study, 41.62% engaged in risky sexual behaviour.[9]

Previous studies in Ethiopia showed that the SRH knowledge of young girls and their SRH service utilisation is very low, which might worsen their SRH problems.[3] A survey conducted in 2019 showed that only 37% of married young girls used family planning (FP) services.[10] On the other hand, one-third of married and 12% of unmarried young girls have an unmet need for FP.[10] Lack of SRH-related information, poor perceptions about SRH, feelings of shame, fear of being seen by others, while they are using the SRH services, restrictive cultural norms and inaccessibility to health services are among the underlying causes for this low service utilisation.[3 11]

Cognizant of these facts, the Ethiopian government launched a Health Extension Programme (HEP) in 2003 as a community-based programme to increase the quality and access to health services in order to prevent disease, promote healthy behaviours, and improve knowledge and service utilisation.[12 13] Youth reproductive health is one of the 18 HEP packages designed to be delivered at a grass-root level.[13] Health extension workers (HEWs) are expected to educate the community on these packages, and specifically, on SRH-related issues, and provide different health services through a home visits, school visits and at health posts (HPs).[14] Nonetheless, to date, to the best of our knowledge, there is limited evidence about the extent to which SRH services are used by young girls.[15] Having such evidence is useful to monitor the performance of HEP in addressing one of the health service uptakes by young girls. This study aims to assess the level of SRH service utilisation of young girls in rural Ethiopia and to examine the role of HEP in this regard.

## METHODS
### Study context
During the data collection, Ethiopia was administratively divided into nine regions and two city administrations. Each region is hierarchically divided into zones, districts and then kebeles. A kebele is the smallest administrative unit expected to serve an average of 5000 people, and a district is the lowest budgeted centre of the government. More than 95% of the kebeles in the country have HPs. HPs are categorised under the primary level of Ethiopian health system tiers, constructed within each kebele and staffed on average by two HEWs. Each HEW is expected to serve 1500–2500 people and there are nearly to 40 000 HEWs in more than 17 000 HPs in Ethiopia.

HEWs are high school graduates with an additional 1 year of education that enables them to be engaged in health education and delivery of FP methods and selected curative services. They are expected to spend 75% of their time in outreach services (home visits, school visits and community campaigns) and 25% on curative services at the HP. For the success of the HEP programme, HEWs are supervised at least weekly and monitored by the nearby primary healthcare unit health professionals.[16] Health education on SRH issues is supposed to be provided by HEWs both at HPs and at the household level. Services like Skilled birth, selected FP methods (intrauterine devices (IUD), Norplant and permanent FP) and treatment for STI or HIV are provided at the health centre or hospital level rather than at the HP level. In these cases, HEWs are expected to refer the clients to the catchment health centre after counselling them.[13]

### Study design, setting and data source
A cross-sectional national survey was conducted using a multistage sampling method to evaluate the Ethiopian rural HEP, and for this study, data were extracted from this survey. The survey covered all the nine administrative regions of the country and data collection was conducted from March to May 2019. In total, 62 (both agrarian and pastoralist regions) districts were covered in the assessment by using appropriate methodological flow. Within each district three kebeles were randomly selected and then within each selected kebeles, 34 households were randomly selected from the total list of households in the kebele. A total of 902 young girls between the ages of 15 and 24 years from a total of 6324 rural households were included in the HEP programme evaluation assessment and we have used these data for analysis. The detailed methods and sampling procedures are described elsewhere.[17]

### Data collection for the HEP assessment
During the HEP programme evaluation assessment, we used a structured questionnaire adapted from different literature.[18–20] Interview-administered field data collection was facilitated using Open Data Kit application with young girls at each kebele. The questionnaire was translated into different local languages of Ethiopia and all study participants were interviewed in their local languages. Before starting the actual data collection, the questionnaire was pretested for all languages out of the actual study area then any language barriers and jargon words were corrected and missed valuable variables were also added to the questionnaire. To minimise recall bias different techniques such as probing questions, and decreasing the study period were used. Data collectors (both male and female) were recruited and given 10-day training on study objectives, data collection methods, ethical procedures and data quality issues.

### Patient and public involvement
No patients involved.

### Measurements
#### Outcome variable
*Young girls' SRH service utilisation*
This is a dichotomous outcome variable that refers to the status of a young girl in using at least one of SRH services: (A) attending health education on SRH issue (during a home visit or at school or during HP visit) or (B) FP

utilisation (utilisation of at least one modern FP method like IUCD, implants, injectable, condoms, pills). The variable assumes a value of '1' if a given young girl used at least one of the SRH services and '0' if she did not use any.

## Key exposure variable
### Exposure to HEW
This is the exposure variable that we used to investigate the role of HEP in enhancing SRH service utilisation of young girls. We measured this variable as follows: young girls who meet HEW for any service at a home visit, school visit or HP visits within a 1-year period prior to data collection time were labelled as 'exposed to HEW' or else they were labelled as 'not exposed'.

## Another independent variable
### Sociodemographic characteristics
Age, educational status, marital status (categorised as never married and ever married (married, separated and widowed)), source of information for FP, region and wealth index of the household (categorised as lower, middle and high).

### Trust in HEW
This is a composite variable constructed from seven perception-related questions, intended to measure a young woman's attitude towards HEWs. Those young girls who answered strongly agree or moderately agree to all seven perception-related questions were considered to have 'good trust in HEWs' and those who answered strongly disagree or moderately disagree were labelled as 'poor trust in HEWs'.

### Knowing the availability of SRH-related HEP services at HP
Young girls who know at least two SRH-related HEP packages/services (health education, delivery, antenatal care, postnatal care, FP, HIV test) were considered as having an 'adequate knowledge' and the other young girls were considered as having 'inadequate knowledge'.

### Modern FP method knowledge
young girls who know at least one modern FP method (ie, injectable, pills, IUD, condom (both male and female) and implants) were labelled as 'knowledgeable about modern FP methods', and those who were not able to list any one of the five methods were considered as 'not knowledgeable about modern FP methods'.

### Knowledge of HIV transmission
This is another composite variable that used responses from nine questions. A young girl who gave at least one correct answer from a total of nine HIV transmission-related questions was labelled as having 'knowledge of HIV transmission' and if she did not give a correct answer for any of the nine HIV transmission-related questions, she was considered as 'not knowledgeable about HIV transmission'.

## Statistical analysis
We used Stata V.14 for the analysis and reported frequencies, percentages, mean and SDs as descriptive summary measures of key variables. We used multilevel binary logistic regression to model the odds of using SRH services. Due to the multistage cluster sampling procedure, individual young girls were nested within kebeles and then within regions. We examined the effect of the individual-level variables, and the region-level variables using two-level logistic regression modelling. During analysis, the characteristics of young girls were taken as individual level (level 1), and regions were taken as level 2. The region was considered as level two to understand regional variability which is useful for future intervention.

Based on this model, the intraclass correlation coefficient (ICC) was calculated to evaluate whether the variation in the odds of using SRH services was primarily within or between regions. Both the crude OR and the adjusted OR (aOR) with their corresponding 95% CI were reported as measures of strength of association.

## RESULTS
### Sociodemographic characteristics
A total of 902 young girls were interviewed, yielding a 100% response rate. The mean age of the respondents was 17.48 years (SD = ±2.27), 816 (80%) young girls were in the age range of 15–19 years, 170 (18.82%) had no formal education and 779 (86.36%) were never married (table 1).

### Knowledge about SRH services and HEP-related perception
Only one hundred and seventeen (12.97%) young girls knew at least one method of FP and 97 (10.75%) young girls were knowledgeable about modes of HIV transmission and prevention. Based on answers of strongly agree or moderately agree to attitude-related questions, 651 (76.77%) young girls had a good perception of the HEW services being friendly to young's FP needs, 602 (71.92) self-reported good trust in HEW and 732 (86.73%) had a good perception of HEP services. Three hundred and forty-nine (38.69%) young girls recommended an increase in the frequency of home visits by HEWs (table 2).

### Utilisation of SRH services
National-level SRH services utilisation of young girls was 19.18% (95% CI 16.74% to 21.89%). Among the 244 young girls that live in pastoralist areas, only 29 (11.90%) used SRH services. Similarly, among 658 young girls that live in agrarian areas, 144 (21.9%) used SRH services. There is a significant amount of variation in the status of SRH utilisation across the region accounting for 17.16% of the total variability ((ICC=17.16%; 95% CI 6.30 to 39.99).

SRH services use among young girls was significantly higher for girls who had exposure to HEP through HEWs' home visits, HEWs' school visits, or during HP

Table 1 Sociodemographic and household-level characteristics of young girls in Ethiopia, 2020

| Characteristics | No | Per cent |
|---|---|---|
| Age | | |
| 15–19 | 721 | 79.9 |
| 20–24 | 181 | 20.1 |
| Educational status | | |
| No formal education | 170 | 18.85 |
| Primary | 514 | 56.98 |
| Secondary and above | 218 | 24.17 |
| Marital Status | | |
| Single | 779 | 86.36 |
| Ever married | 123 | 13.64 |
| Region | | |
| Tigray | 117 | 12.97 |
| Afar | 63 | 6.98 |
| Amhara | 169 | 18.74 |
| Oromia | 162 | 17.96 |
| Somali | 97 | 10.75 |
| Benishangul Gumuz | 50 | 5.54 |
| SNNPR | 150 | 16.63 |
| Gambela | 36 | 3.99 |
| Harari | 58 | 6.43 |
| Source of information for FP | | |
| HEW | 186 | 23.4 |
| Other health professionals | 52 | 6.54 |
| Other sources (TV, radio, friend, relatives and printed materials) | 557 | 70.06 |
| Wealth index | | |
| Lower | 290 | 32.88 |
| Middle | 186 | 21.09 |
| Higher | 406 | 46.03 |

FP, family planning; HEW, health extension worker; SNNPR, South Nation Nationality People Region; TV, television.

Table 2 Knowledge and perception of young girls on sexual and reproductive health-related health extension package in Ethiopia, 2020

| Variables | No | Per cent |
|---|---|---|
| Knowledge of FP methods | | |
| Knowledgeable | 117 | 12.97 |
| Not knowledgeable | 785 | 87.03 |
| Comprehensive knowledge about HIV | | |
| Knowledgeable | 97 | 10.75 |
| Not knowledgeable | 805 | 89.25 |
| Have a good perception of the HEW code of conduct | | |
| Strongly disagree | 41 | 4.83 |
| Moderately disagree | 156 | 18.4 |
| Moderately agree | 490 | 57.78 |
| Strongly agree | 161 | 18.99 |
| Have good trust in HEW | | |
| Strongly disagree | 36 | 4.3 |
| Moderately disagree | 199 | 23.78 |
| Moderately agree | 438 | 52.33 |
| Strongly agree | 164 | 19.59 |
| Good perception on SRH service delivery by HEWs | | |
| Strongly disagree | 31 | 3.67 |
| Moderately disagree | 81 | 9.6 |
| Moderately agree | 442 | 52.37 |
| Strongly agree | 290 | 34.36 |
| Knowing the availability of SRH-related HEP services at HP | | |
| Health education | 633 | 71.08 |
| Family planning | 540 | 60.39 |
| Antenatal care | 534 | 58.82 |
| Delivery service | 269 | 30.49 |
| Postnatal care | 359 | 40.39 |
| Allocation of time for HP and home visit | | |
| Increase the frequency of home visits | 349 | 38.69 |
| Increasing the time, they spend at HP | 131 | 14.52 |
| Increase both of them | 347 | 38.47 |
| Keep the current allocation of time | 75 | 8.31 |

FP, family planning; HEW, health extension worker; HP, health post; SRH, sexual and reproductive health.

visits (70.5%) as compared with girls who did not have such exposure (37.6%).

### Determinants of SrH services utilisation use among young girls

Young girls who were exposed to HEW were three times more likely to use SRH services than girls who were not exposed to HEW (aOR 3.13, 95% CI 2.03 to 4.85).

Other factors that are associated with increased SRH services utilisation are knowing SRH-related services are available at HPs (aOR 3.05, 95% CI 1.75 to 5.33), having good trust in HEW (aOR 1.82, 95% CI 1.07 to 3.10), being in the age range of 20–24 years (aOR 2.93, 95% CI 1.86 to 4.63) and being ever married (aOR 6.54, 95% CI 3.82 to 11.19) compared with their counterparts (table 3).

### DISCUSSION

In this study, the status of SRH service utilisation of young girls was very low, however, it was increased when they were exposed to HEP through health education provided by HEW. Exposure to HEWs, knowing about HEP service availability, and having trust in HEW were significantly associated with increased SRH service utilisation among

**Table 3** Individual and region level factors associated with young girls' SRH service utilisation from Hep in Ethiopia, 2020

| Variables | Utilisation of SRH | | cOR (95% CI) | aOR (95% CI) |
| --- | --- | --- | --- | --- |
| | No | Yes | | |
| Exposure to HEW | | | | |
| Not exposed | 62.41 | 29.48 | 1.00 | 1.00 |
| Exposed | 37.59 | 70.52 | 3.61 (2.47 to 5.26) | 3.13 (2.03 to 4.85)* |
| Understanding HEP service availability | | | | |
| Inadequate | 91.5 | 8.5 | 1.00 | 1.00 |
| Adequate | 75.66 | 24.34 | 3.53 (2.19 to 5.69) | 3.05 (1.75 to 5.33) * |
| Trust in HEW | | | | |
| Good | 87.66 | 12.34 | 2.17 (1.38 to 3.42) | 1.82 (1.07 to 3.10)* |
| Poor | 77.08 | 22.92 | 1.00 | 1.00 |
| Age | | | | |
| 15–19 | 86.13 | 13.87 | 1.00 | 1.00 |
| 20–24 | 59.67 | 40.33 | 4.44 (3.01 to 6.55) | 2.93 (1.86 to 4.63)* |
| Marital status | | | | |
| Ever married | 86.39 | 13.61 | 8.24 (5.17 to 13.13) | 6.54 (3.82 to 11.19)* |
| Never married | 45.53 | 54.47 | 1.00 | 1.00 |

*P value less than 0.05.
HEP, Health Extension Programme; HEW, health extension worker; SRH, sexual and reproductive health.

young girls. While older age and being ever married also were significantly associated with increased SRH service utilisation, educational status of young girls, wealth index of the household, perception about SRH service delivery by HEW, and perception of the youth-friendliness of HEWs were not significantly associated with SRH service utilisation of young girls.

SRH service utilisation of young girls was relatively lower compared with the findings of previous studies in Ethiopia.[20–23] One possible reason for the discrepancy might be the difference in the study settings; participants in the current study are from rural areas where resources and access to services are very limited.[23] Another study also explained that this low utilisation might also be linked to a low level of knowledge among young girls implying that the focus given to young girls in rural areas by HEWs is likely to be minimal.[24] This assumption is also supported by the low percentage of young girls in the current study having exposure to HEP services during a home visit or during their schooling. Encouraging HEWs to give due attention to young girls in rural areas might be one of the strategies that should be followed to increase these girls' service uptake in rural Ethiopia, which is the home of the significant majority of the population, which implies that investing out there can improve the country in general.[13]

One of the HEP packages was designed to increase youth SRH service utilisation through health education during home visits, school visits and HP-based service provision.[24 25] In line with this, this study findings showed that HEWs had a significant role in increasing the SRH service utilisation of young girls. Based on the 2015 Ethiopian second generation HEP plan, whenever the HEWs meet young girls in their home, at school, or HP, the odds of young girls' SRH service utilisation were increased.[13] This is a general indication that strengthening the role of the HEWs in implementing HEP strategies can improve the SRH service utilisation of young girls.

Adequate knowledge of available components of HEP is also significantly associated with increased service utilisation among young girls. Among different components of SRH-related HEP packages, most of the respondents recalled health education and FP services. This may reflect the differential focus given to FP education by HEWs during service delivery. Studies in Ethiopia and Uganda showed that knowing the place of SRH service increases service utilisation.[25 26] One of the global-quality standard measures also indicates that someone is knowledgeable about their health only if they know where and when to get those services.[27] This is based on the assumption that the more they know about the available services the more likely it is for them to use the service in the nearby HP.

Confidentiality is one of the major ethical codes in medicine and SRH service utilisation requires a similar level of confidentiality because of its sensitivity.[11 28] Young girls are very worried about confidentiality, and they do not want to be seen by their parents, friends, or other people they know (relatives, teachers or neighbours) while receiving SRH services because this knowledge makes them vulnerable to stigma and mortification.[29] The International Federation of Gynecology and Obstetrics recognises that health professionals are obligated to adhere to their patients' confidentiality.[30] In this study, young girls who have a good trust in HEWs were more likely to use SRH services than their counterparts. Other studies also reported that increasing the privacy and confidentiality of youths improved their service utilisation.[31 32] Young girls

should be provided confidential and flexible service in a youth friendly manner which can increase their trust and service utilisation.[33 34]

In line with a previous Ethiopian study, conducted at Baher Dar, young girls between the ages of 20 and 24 years were more likely to use SRH services than girls aged 15–19 years.[32] SRH demand increases with age due both to a change in marital status that could lead to an interest in birth spacing, and increased educational status that could increase their ability to understand the risk of exposure to sexual practice.[8]

Although the findings of the Ethiopian Demographic Health Survey showed that many young girls are engaged in sexual intercourse before marriage, the issue of SRH is culturally sensitive and socially unacceptable for unmarried girls which may decrease service utilisation among young girls.[8] In agreement with this, unmarried young girls in the current study were less likely to use SRH services from HEP. The negative attitude of HEWs towards the use of SRH services by unmarried young girls, which is mainly due to lack of training, was provided as a justification for similar findings in a previous Ethiopian study.[35]

Some limitations of this study include the survey's failure to assess the attitude of HEWs about SRH delivery to unmarried young girls which, as just discussed, likely makes young girls less likely to use SRH services from HEP. Other limitations include recall bias on the part of young girls surveyed which might have underestimated the magnitude of SRH services used by these girls. Further, the survey used both female and male data collectors; using male data collectors for this kind of study may have affected the magnitude of SRH service utilisation reported. This study also did not ask about barriers that hinder HEWs from increasing youth service utilisation. Additionally, we did not perform a qualitative study to assess the effect of cultural barriers such as existing norms, beliefs and gender-related barriers on SRH service using. However, using nationally representative data is one of the important strengths of this study.

## CONCLUSION

This study demonstrates that SRH service utilisation of young girls in rural Ethiopia is very low. Our findings also indicate that HEWs have a significant contribution to improving SRH service utilisation of young girls through health education during home visits, during school visits and at HPs. However, more effort might be required to significantly increase the exposure of young girls to HEP services to increase service uptake. Encouraging young girls to build trust in HEWs and make young girls know what SRH-related services are available at the HP is the first step towards improving service uptake. HEWs need to give special focus on younger girls and never married young girls to increase their service uptake.

**Author affiliations**
¹School of Public Health, Hawassa University College of Medicine and Health Sciences, Hawassa, Ethiopia
²Department of Statistics, College of Natural and compultional Sciences Addis Ababa University, Addis Ababa, Ethiopia
³Aklilu Lemma Institute of Pathology, Addis Ababa University, Addis Ababa, Ethiopia
⁴MERQ Consultancy PLC, Addis Ababa, Ethiopia
⁵Federal Ministry of Health, Addis Ababa, Ethiopia
⁶Institute of Health Science, Jimma University, Jimma, Ethiopia

**Acknowledgements** We are very thankful to MERQ consultancy for granting us to access this data source for this study. We acknowledge Ms. Saron who thoroughly edited this manuscript for language usage, spelling and grammar. Pre-Publication Support Service (PREPSS) supported the development of this manuscript by providing prepublication peer-review and copy editing.

**Contributors** AMT YK GM TLD contributed to the conceptualization, designing, and conduct of the national HEP assessment, MGJ TLD MJ YS conceptualized the idea coved within this paper, MJ MB led the data analysis and was involved in the drafting of the MS. GM MGJ TLD supervised the write-up of the draft, all coauthors critically reviewed the drat MS and contributed intellectual inputs to improve the content of the MS and endorsed the final version of the MS to be submitted for possible publication.

**Funding** The study was conducted by MERQ Consultancy PLC as part of a national assessment on the Health Extension Programme of Ethiopia. The national assessment was financed through a grant from the Bill & Melinda Gates Foundation (INV010174).

**Disclaimer** The funder provided financial support to MERQ that covered professional fees for leaders but did not have any additional role in the study design, data collection and analysis, decision to publish, or preparation of the manuscript. The specific roles of these authors are articulated in the 'author contributions' section.

**Competing interests** None declared.

**Patient and public involvement** Patients and/or the public were not involved in the design, or conduct, or reporting, or dissemination plans of this research.

**Patient consent for publication** Not applicable.

**Ethics approval** This study involves human participants and was approved by Ethiopian public health association (EPHI)-1613/820. Participants gave informed consent to participate in the study before taking part.

**Provenance and peer review** Not commissioned; externally peer reviewed.

**Data availability statement** Data are available on reasonable request. We will share the data upon the request Name: GM (Phd)E-mail: girmay.m@ merqconsultancy.org.

**ORCID iDs**
Meskerem Jisso http://orcid.org/0000-0003-0102-658X
Merga Belina Feyasa http://orcid.org/0000-0001-5862-5941
Tegene Legese Dadi http://orcid.org/0000-0003-1361-9649

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
