## [Reviewer comments · BMJ Open]

ARTICLE DETAILS

TITLE (PROVISIONAL)	Sexual and reproductive health service utilization of young girls in rural Ethiopia: What are the roles of health extension workers?: Community based cross-sectional study
AUTHORS	Jisso, Meskerem; Feyasa, Merga; Medhin, Girmay; Dadi, Tegene; Simachew, Yilkal; Denberu, Bisrat; Jebena, Mulusew; Alemayehun, Yibeltal; Teklu, Alula M.

VERSION 1 – REVIEW

REVIEWER	Metusela, Christine University of Wollongong, School of Medicine
REVIEW RETURNED	27-Nov-2021

GENERAL COMMENTS	This is an important topic but before being accepted for publishing it needs comprehensive editing. In the title and all the way through the manuscript the expression 'youth girls' is used. This is an incorrect use of the term. This will need to be corrected to "young girls" or "female youth". Page 4 line 52-57: the strengths and limitations should be in the discussion section and written out in full sentences, not in dot form. Page 4 line 63: what is meant by "share of youth"? This doesn't make sense. Page 4 lines 67 and line 68 need rewording to make the meaning more clear. Page 7 line 136: "We extracted a total of 902 youth girls". What does this mean? This needs to be explained more clearly. Are you meaning that 902 young girls were interviewed or that 902 young girls were selected from a database? Page 8 line 172 needs clarification as it appears to be saying that girls that did not get the correct answer for all the nine ways of HIV transmission were labelled as not knowledgeable about HIV transmission. However this does not make sense and I think the authors are meaning that girls that did not get the correct answer of any of the nine way of HIV transmission were labelled as not knowledgeable about HIV transmission. Page 9 line 189. This ethical process needs to be written more clearly. It says the ethical letter was obtained. Do the authors mean that ethics letter of approval? Page 9 line 195: this sentence needs to be written in a consistent style – change the one-fifth to numbers. Page 10. The results section needs to be more clear about what findings were statistically significant. Page 13 line 252: reword HEWS had a great role – do you mean an important role, a significant role?
--

	Page 13 line 267 needs to be reworded. The sentence about young girls being very worried needs to connect clearly with what they are very worried about. Page 14 lines 277-288: These two paragraphs need to be re-written to be clear about what each paragraph is saying. -Many of the articles that are cited are greater than ten years old. It would be good to include some more recent literature. Typos and grammatical errors Grammar throughout the manuscript needs to be checked as there are inconsistencies with past tense/present tense. The whole manuscript will need to be thoroughly edited for typos and grammatical and formatting errors. I have listed some below, but this list is not exhaustive: Page 3 line 33: nine hundred two should be nine hundred and two. Page 3 line 49: on this line should be along this line. Page 4 line 57: "recall bias that might have been" should be "recall bias might have been". Page 4 line 62: "and majority of them" should be "and the majority of them". Page 4 line 72: there are two commas after the word Ethiopia. Page 5 line 91: A space between HEP and in is needed. Page 5 line 95: the word divide should be divided. Page 5 line 96: The word hieratically is used wrongly. I think the authors are meaning hierarchically. Page 5 line 99: A full stop is missing between health posts and Health posts. Also be consistent with capitalisation e.g. this line has both lower and capital letters for health posts. Page 5 line 102: Full stop missing after Ethiopia. Page 6 line 120: "each of the selected kebele" should be "each selected kebele". Page 6 line 121: "whose age is 15-24" should be "between the age of 15-24". Page 6 line 122: "procedures was described" should be "procedures are described" Page 6 line 130: "by their local languages" should be "in their local languages". Also "Then the questionnaire" doesn't read well like that. The word "then" should be removed so that it says: "The questionnaire". Page 6 line 131: barrios typo should be "barriers" Page 7 line 149: "within one year period" should be "within a one-year period". Page 7 lines 157-160 use the phrase "trust on HEW" which should be "trust in HEW". Page 7 line 160. There is a full stop missing. Page 8 line 165: "one of modern FP methods" should be "one modern FP method". Page 8 line 169: "variable that have used" should be "variable that used". Page 8 lines 179: "using a two-level logistic regression modelling" should be "using two-level logistic regression modelling". Page 8 line 187: A full stop is missing. Page 9 line 191: "Result" should be "Results". Page 10 line 202: "One hundred seventeen" should be "One hundred and seventeen". Page 10 line 204: "Six hundred fifty-one" should be "six hundred and fifty-one". Page 11 line 213: Full stop missing. Page 12 line 232 "and it was increased" would read better saying "however it was increased".
--	--

	Page 13 line 251 “school visit” should be either “school visits” or “school visitation”. Page 13 line 252: “current finding” should be “current findings”. Page 13 line 260: “during any service delivery” should be “during service delivery”. Page 13 line 264: “the available service” should be “available services”. Page 14 line 272: HEW should be HEWs. Page 14 line 273: service should be services. Page 13 line 277: “In line with previous...” should be “In line with a previous”. Formatting The formatting also needs to be corrected and made consistent.  -The spacing between paragraphs is inconsistent. -Sometimes there are spaces between brackets and sometimes not. -Sometimes there are double spaces between words and between sentences.
--	---

REVIEWER	Jonas, Kim South African Medical Research Council, Health Systems Research
REVIEW RETURNED	24-Mar-2022

GENERAL COMMENTS	This is a great paper, well-written paper. I really enjoyed reading the paper. there are some minor editorial things that the authors may want to clean up... see below line 63: place space between parts and the bracket... line 72: double commas after Ethiopia- please delete one. line 91: add space between HEP and in... line 117: space before the bracket line 123: space before the bracket line 127: space before the bracket general comment- please check your spacing in the entire document.
--

VERSION 1 – AUTHOR RESPONSE

Reviewer-1

We preferred to use young girls, and the whole document was edited accordingly

Page 4 line 52-57: we couldn't amend this because it is part of submission criteria for BMJ open journal.

Page 4 line 63: we want to explain the proportion of youth was high but now It is corrected

Page 7 line 136: It is corrected

Page 8 line 172: It is more clarified and corrected

Page 9 line 189. It is corrected

Page 9 line 195: It is corrected

Page 10: We have tried to edit the whole English writeup and all significant variables were written clearly.

Page 13 line 252: It is corrected (significant role)

Page 13 line 267: It is corrected

Page 14 lines 277-288: We tried to reword the whole manuscript

We have tried to update reference and replaced some old references.

Copy editing was done throughout this manuscript and all grammatical problems were amended.

Page 3 line 33: It is corrected

Page 4 line 57: This was originally written as a limitation in the abstract section, but after fixing the word, it was moved to the discussion section.

Page 4 line 62: It is corrected

Page 4 line 72: It is corrected

Page 5 line 91: It is corrected

Page 5 line 95: It is corrected

Page 5 line 96: It is corrected

Page 5 line 99: It is corrected

Page 5 line 102: It is corrected

Page 6 line 120: It is corrected

Page 6 line 121: It is corrected

Page 6 line 122: It is corrected

Page 6 line 130: It is corrected

Page 6 line 131: It is corrected

Page 7 line 149: It is corrected

Page 7 lines 157-160: It is corrected

Page 7 line 160: It is corrected

Page 8 line 165: It is corrected

Page 8 line 169: It is corrected

Page 8 lines 179: It is corrected

Page 8 line 187: It is corrected

Page 9 line 191: It is corrected

Page 10 line 202: It is corrected

Page 10 line 204: It is corrected

Page 11 line 213: It is corrected

Page 12 line 232: It is corrected

Page 13 line 251: It is corrected

Page 13 line 252: It is corrected

Page 13 line 260: It is corrected

Page 13 line 264: It is corrected

Page 14 line 272: It is corrected

Page 14 line 273: It is corrected

Page 13 line 277: It is corrected

All formatting issues were amended

Reviewer 2

Thank you sir for your support.

line 63: It is corrected

line 72: It is corrected

line 91: It is corrected

line 117: It is corrected

line 123: It is corrected

line 127: It is corrected

VERSION 2 – REVIEW

REVIEWER	Metusela, Christine University of Wollongong, School of Medicine
REVIEW RETURNED	20-Jun-2022

GENERAL COMMENTS	The manuscript has been much improved with your revisions. However, there are still some edits to make: Line 135 says “different techniques were used to minimize recall bias. “ -What techniques? Describe these. Line 237 words are missing in the following sentence: “In this study, the status of SRH service utilization of young girls was very however...” Line 282 The following sentence needs correcting: “Young girls should be provided confidential and flexible service in youth a youth friendly manner” -it seems like the first use of the word “youth” needs to be deleted. Line 304 The following sentence needs correcting: “such as existing norms, beliefs, gender related barriers and etc. on” should say something like “...and gender related barriers on...” Page 12 of 42 Table 3: It is unclear what the p-values are. Other edits: Line 33 should be “included in” NOT included to Line 55 should be “may” NOT that might Line 56 “could not” NOT couldn’t Line 63 should say “(79% male and 78% female)” NOT (79% of males and 78% of females) Line 74 full stop in the wrong place. Line 72 tenses are in the past so “underwent” NOT undergo Line 74 delete “in the world” and at the beginning of the sentence (Line 72) begin with “Globally...” Line 75 delete the word “are” Line 81 should say “feelings of shame” NOT feeling of shame Line 90 should say “through home visits” NOT through a home visits Line 107 should say “with an additional” NOT with additional Line 139 should say “no patients involved” NOT no patient involved Line 190 needs a full stop Line 192 use capital letters for: “Ethiopian public health association” Table 2 page 10 of 42 should say “Good perception of” NOT Have a good perception of Line 284 should say “between the age” NOT at the age Line 293 says SHR but the text mostly uses SRH. Check for consistency.
--

VERSION 2 – AUTHOR RESPONSE

Reviewer-1

Line 135: It is described in detail

Line 237: It was corrected

Line 282: It was corrected

Line 304: It was corrected

Page 12 of 42 Table 3: It was clearly stated and corrected accordingly

Line 33: It was corrected

Line 55: It was corrected

Line 56: It was corrected

Line 63: It was corrected

Line 74: It was corrected

Line 72: It was corrected

Line 74: It was corrected

Line 75: It was corrected

Line 81: It was corrected

Line 90: It was corrected

Line 107: It was corrected

Line 139: It was corrected

Line 190: It was corrected

Line 192: It was corrected

Table 2 page 10 of 42: It was corrected

Line 284: It was corrected

Line 293: It was corrected

Reviewer comments	Response
Line 135 says "different techniques were used to minimize recall bias. " -What techniques? Describe these.	It was described in detail
Line 237 words are missing in the following sentence: "In this study, the status of SRH service utilization of young girls was very however..."	It was corrected
Line 282 The following sentence needs correcting: "Young girls should be provided confidential and flexible service in youth a youth friendly manner" -it seems like the first use of the word "youth" needs to be deleted.	It was corrected

Line 304 The following sentence needs correcting: “such as existing norms, beliefs, gender related barriers and etc. on” should say something like “...and gender related barriers on...”	It was corrected
Page 12 of 42 Table 3: It is unclear what the p-values are.	It is clearly stated and corrected accordingly
Line 33 should be “included in” NOT included to	It was corrected
Line 55 should be “may” NOT that might	It was corrected
Line 56 “could not” NOT couldn’t	It was corrected
Line 63 should say “(79% male and 78% female)” NOT (79% of males and 78% of females)	It was corrected
Line 74 full stop in the wrong place.	It was corrected
Line 72 tenses are in the past so “underwent” NOT undergo	It was corrected
Line 74 delete “in the world” and at the beginning of the sentence (Line 72) begin with “Globally...”	It was corrected
Line 75 delete the word “are”	It was corrected
Line 81 should say “feelings of shame” NOT feeling of shame	It was corrected
Line 90 should say “through home visits” NOT through a home visits	It was corrected
Line 107 should say “with an additional” NOT with additional	It was corrected
Line 139 should say “no patients involved” NOT no patient involved	It was corrected
Line 190 needs a full stop	It was corrected
Line 192 use capital letters for: “Ethiopian public health association”	It was corrected
Table 2 page 10 of 42 should say “Good perception of” NOT Have a good perception of	It was corrected
Line 284 should say “between the age” NOT at the age	It was corrected

Line 293 says SHR but the text mostly uses SRH. Check for consistency.	It was corrected
--	------------------